# The Influence of Extractant Composition on the Asphaltenes Extracted from Asphalt

Dachuan Sun [1,2,*,†] , Yang Song [2,*,†] and Feiyong Chen [2]

1   School of Transportation Engineering, Shandong Jianzhu University, Jinan 250101, China
2   Huzhou Nanxun District Jianda Ecological Environment Innovation Center, Shandong Jianzhu University, Jinan 250101, China
*   Correspondence: sundc@iccas.ac.cn (D.S.); songyang20@sdjzu.edu.cn (Y.S.); Tel.: +86-199-5312-0288 (D.S.)
†   These authors contributed equally to this work.

**Abstract:** The compositions of extractants containing xylene (G) and n-heptane (P) recovered and reused in the asphaltene extraction process were determined by densimetry and the extracted asphaltenes were analyzed to study the influence of extractant composition on their elemental, spectral and structural properties. With increasing the number of extraction, the G/P ratio in extractant increases, which dissolves more asphaltenes of higher aromaticity and causes a decrease of asphaltene yield, the polarity and aromaticity in molecular structure of the extracted asphaltenes. Asphaltenes extracted at a higher G/P ratio were found to have less fractions of N and O elements as well as higher fractions of H and S elements, a lower C/H atom ratio and molecular unsaturation. Moreover, they have a smaller fraction of aromatic rings and aromatic carbon ratio, a higher substitution rate on aromatic rings, a higher fraction of alkyl chain and free ends in their average molecular structure. Compared with the base asphalt, asphaltenes' infrared absorptions generally move towards smaller wave numbers owing to more aromaticity in their molecules and subsequent stronger conjugative and inductive effects. The asphaltenes extracted at a higher G/P ratio have a denser packing of alkyl chains and a looser packing of aromatic rings, according to their spectra of X-ray diffraction.

**Keywords:** densimetry method; extractant composition; n-heptane and xylene; asphaltene extraction; molecular structure

## 1. Introduction

Asphalt is one kind of coating materials widely used for roof waterproofing, interface bonding and surface layers of high-grade highways [1,2]. Asphalt has a complex chemical structure and composition [3–5]. According to the different solubility, it can be divided into four components: saturate, aromatic, resin and asphaltene [1,5,6]. In our previous work [7], using the solubility discrepancy of asphaltene in different solvents [8], a simple scheme for extracting asphaltenes from asphalt at room temperature was proposed and verified experimentally. The scheme firstly dissolves the asphalt in a good solvent (denoted as G) and then uses a poor solvent (denoted as P) to precipitate the asphaltenes out. As the scheme is carried out at room temperature without heating, it avoids the problems of high energy cost, expensive devices, and safety risks of flammable steam in traditional methods (JTG E20-2011 or IP 143) of asphaltene extraction [6,9]. After extraction, the good solvent was mixed with the poor solvent. To reduce costs, these waste (used) solvents need to be recovered and reused as new extractants. However, the yield decreased significantly when the recovered solvent was used, probably caused by the variance of extractant composition. To better study the effect of extractant composition on asphaltene yield, a method is needed to analyze the extractant concentration or G/P ratio, which is also helpful to optimize the separation condition and improve the separation efficiency of the waste extractants.

Alkanes and aromatics are volatile components in coating materials and widely used as extractants or eluents [3,10]. When extracting asphaltenes from asphalt, the good

solvents are generally aromatic hydrocarbons, such as xylene and toluene while the poor solvents are n-alkanes, such as n-heptane and n-hexane [3,5,9]. Xylene is a $C_8$ aromatic mixture containing isomers and a small amount of ethylbenzene [11,12]. Both xylene and heptane are volatile and miscible with each other in any ratio. Heptane is a poor solvent or precipitant for asphaltenes, owing to the difference of molecular structures (thus higher enthalpy penalty) between heptane and asphaltenes as well as high molecular weight of asphaltene (thus less mixing entropy) [3,6]. Asphaltene is precipitated only when the G/P ratio is below a critical precipitation point [9,13,14]. The analysis of G/P ratio in extractant is helpful to judge whether the critical precipitation point is exceeded, and helpful to guide the industrial extraction of asphaltenes. Compared with the analytical methods such as high performance liquid chromatography (HPLC), nuclear magnetic resonance (NMR) and gas chromatography-mass spectrometry (GC-MS) [15,16], the densimetry method may be the simplest and cheapest without requiring expensive equipment [17].

The physical properties [2,18] and molecular structures [4,5] of the extracted asphaltenes may vary with extraction number or G/P ratio. Asphaltenes have been analyzed via elemental analysis (EA), NMR, Fourier transform infrared (FTIR) and X-ray diffraction (XRD) methods [19,20] to get average molecular structure [3] to compare with other bitumen components [6] or with asphaltenes precipitated from different alkanes [10,21]. There are few studies for the effect of G/P ratio on the properties of extracted asphaltene. Such a study is helpful to evaluate the property discrepancy and to judge whether these asphaltenes can be mixed and used together. In this paper, the compositions of recovered extractants were analyzed via densimetry, while the properties of asphaltenes extracted at different G/P compositions were studied. Mechanisms were discussed for the variance of asphaltene properties and for the decrease of asphaltene yield with extraction number or G/P ratio.

## 2. Materials and Methods

### 2.1. Materials and Devices

The asphalt used is Qilu No. 70 base asphalt from Qilu Petrochemical Company (Zibo, China). Potassium bromide (KBr, crystals, of 99% purity) with chemical abstracts service number (CAS No.) 7758-02-3 is purchased from Merck & Co., Ltd. (Beijing, China). Deuterated chloroform (CDCl$_3$, CAS No. 865-49-6, 99.8% purity) containing 0.03% $v/v$ tetramethylsilane (TMS) is purchased from Cambridge Isotope Laboratories, Inc. (Beijing, China). Xylene (xylene isomer + ethylbenzene, CAS No. 1330-20-7) and n-heptane (CAS No. 142-82-5) of analytical reagent (AR) grade are purchased from Macklin Biochemical Technology Company (Shanghai, China).

The glasswares were purchased from Chongqing Synthware Glass Company (Chongqing, China). The HJ-6A thermostatic magnetic stirrer is purchased from Jiangsu Zhongda Technology Instrument Company (Nanjing, China). The PX224ZH/E electronic balance and STX portable balance are purchased from Ohaus International Trading (Shanghai) Company (Shanghai, China). The sand-core filter device with a filter bottle of 1000 mL volume and a filter cup of 300 mL volume is purchased from Jiangsu Feida Glass Products Company (Nanjing, China) and used to get the precipitated asphaltenes by suction filtration. The RE-52AA rotary evaporator and SHZIII circulating water vacuum pump are purchased from Shanghai Yarong Biochemical Instrument Factory (Shanghai, China) and used to recover extractant via vacuum distillation. The medium-speed setting filter paper is purchased from Hangzhou Fuyang Northwood Pulp Company (Hangzhou, China). The 0.45 μm filter membrane is purchased from Shanghai Xinya Purification Device Factory (Shanghai, China). The polypropylene dropper and other consumables are supplied by Shandong Laboratory Experimental Instrument Company (Jinan, China).

*2.2. Experimental Procedures*

2.2.1. Preparation of Standard Samples of Extractants

The subscripts P and G represent poor solvent (n-heptane) and good solvent (xylene), respectively, in Table 1 and in the following. All operations were carried out in a fume hood. The room temperature was maintained at 20 °C by an air conditioner.

**Table 1.** The volume fraction $v$ and mass fraction $w$ of each component for standard samples of extractants containing xylene (G) and n-heptane (P) at room temperature.

| No. | P (mL) | G (mL) | Sum (mL) | $v_P$ | $v_G$ | $w_P$ | $w_G$ |
|-----|--------|--------|----------|-------|-------|-------|-------|
| 0 | 0 | 20 | 20 | 0% | 100% | 0.00 | 1.00 |
| 1 | 2 | 18 | 20 | 10% | 90% | 0.08 | 0.92 |
| 2 | 4 | 16 | 20 | 20% | 80% | 0.17 | 0.83 |
| 3 | 6 | 14 | 20 | 30% | 70% | 0.25 | 0.75 |
| 4 | 8 | 12 | 20 | 40% | 60% | 0.35 | 0.65 |
| 5 | 10 | 10 | 20 | 50% | 50% | 0.44 | 0.56 |
| 6 | 12 | 8 | 20 | 60% | 40% | 0.54 | 0.46 |
| 7 | 14 | 6 | 20 | 70% | 30% | 0.65 | 0.35 |
| 8 | 16 | 4 | 20 | 80% | 20% | 0.76 | 0.24 |
| 9 | 18 | 2 | 20 | 90% | 10% | 0.88 | 0.12 |
| 10 | 20 | 0 | 20 | 100% | 0% | 1.00 | 0.00 |

The procedures for preparing standard samples are as follows:

1. 30 mL cylindrical bottles are cleaned, dried and put on the desk. The bottles and their caps are marked with numbers from 0 to 10 by an oil-based pen;
2. The n-heptane is pipetted into graduated cylinders until a target volume is reached;
3. The n-heptane is poured into a glass bottle. The bottle number and corresponding volume of n-heptane for each sample are listed in Table 1;
4. The xylene is pipetted into another graduated cylinders until its target volume is reached. Then it is poured into the glass bottle with the designated number. The bottle number and volume of xylene are shown in Table 1;
5. The bottles are sealed and shaken for 1 min to mix sufficiently.

The sum of volume is 20 mL for each sample, as shown in Table 1. To simplify the subsequent analysis, the volume fraction $v$ and mass fraction $w$ of each component are calculated and listed in Table 1, according to the quantities [11,12] in Table 2. The graduated cylinders were calibrated by weighing 10 mL of ultrapure water on the PX224ZH/E balance and the density $\rho$ of water at 20 °C is 0.9982 g/mL.

**Table 2.** Basic physical quantities of n-heptane (P) and xylene (G) at room temperature.

| Physical Quantity | P | G | Physical Meaning |
|-------------------|---|---|------------------|
| $\rho$ | 0.68 g/mL | 0.86 g/mL | Density |
| $M_W$ | 100.20 g/mol | 106.17 g/mol | Molecular weight |
| $T_b$ | 98 °C | 140 °C | Boiling point |

2.2.2. Procedures to Get Extracted Asphaltene and Real Samples of Extractants

A detailed method of recovering solvents and extracting asphaltenes from bitumen was documented in a previous report [7]. A total of 3 consecutive extractions were performed and the main steps are as follows:

1. In the first extraction, a total of 5 g asphalt is dissolved in 50 mL xylene of AR purity and then poured into 500 mL n-heptane of AR grade to precipitate the asphaltenes. The solution is then filtered to get the filtrate and filter cake, respectively. The filter cake is dried to obtain asphaltenes (denoted as A1). The weight ratio of A1 to asphalt is defined as asphaltene yield;

2. The above filtrate is recovered via vacuum distillation and the vacuum is provided by connecting SHZIII vacuum pump. The continuously supplied tap water (around 15 °C) is used for condensation and different distillates are obtained by controlling the temperature of the water bath of the rotary distillation equipment. The temperature firstly rises from room temperature to 45 °C and is then held for 1 h, during which the dripping of condensed liquid from the condenser is observable from 30 °C and usually completed within 30 min. The collected fraction is sealed and denoted as P1. Then, the temperature is raised from 45 °C to 80 °C and held for 1.5 h. The dripping of condensed liquid starts from 60 °C and is usually finished within 45 min. The collected fraction is sealed in another bottle and denoted as G1;

3. In the second extraction, asphalt is dissolved in G1, and then poured into P1 to precipitate the asphaltenes (A2). The ratio of asphalt/G1/P1 is maintained at 5 g/50 mL/500 mL for consistency. Subsequent operations are the same as steps (1) and (2), except that the collected condensates at 45 °C (heptane-rich fraction) and 80 °C (xylene-rich fraction) are denoted as P2 and G2, respectively;

4. In the third extraction, asphalt is dissolved in G2, and then poured into P2 to precipitate the asphaltenes (A3). The ratio of asphalt/G2/P2 is maintained at 5 g/50 mL/500 mL for comparison. Subsequent operations are the same as steps (1) and (2). The collected fractions at 45 °C and 80 °C are denoted as P3 and G3, respectively.

The concentration of real samples obtained above (P1, P2, P3 and G1, G2, G3) was measured by densimetry. The elements, spectra and structures are analyzed for the extracted asphaltenes (A1, A2, and A3).

### 2.2.3. Procedures of the Densimetry Method

A fitting curve or equation is made based on the standard samples and then used to analyze real samples. The main steps are as follows:

1. From the standard samples prepared according to Table 1, a 10 mL liquid is transferred to the graduated cylinder with a pipette for each sample;

2. The graduated cylinder is weighed on the PX224ZH/E electronic balance to calculate the mass and density of the sample. Each sample is independently pipetted and measured four times, and the resulting densities are averaged as the final density;

3. A standard curve of densimetry is obtained by plotting the density versus the sample concentrations. The curve is fitted to obtain the standard equation via ORIGIN software (version: OriginPro 2018C 64-bit SR1 for Windows);

4. For the real samples recovered by rotary evaporation, their densities are measured according to the same procedures as the standard samples. The concentration is obtained by substituting the measured density into the equation.

As the blank signals ($Y_B$) are chosen as the starting points of standard curves, the limits of detections (LOD) are three standard deviations of the blank ($s_B$). Seven replicates are analyzed for blank samples at $v_G = 0$ to calculate the standard deviations and the LOD of the densimetry method.

### 2.3. Characterization Methods

For extracted asphaltenes (A1, A2, and A3), their elements, spectra and structures were analyzed via EA, NMR, FTIR and XRD methods. The same characterization was performed on the base bitumen for comparison.

EA was performed via the Elemantar Vario EL cube Elemental Analyzer for the C, H, N and S elements (CHNS mode) at 25 °C. Firstly, a standard sample of 2 mg of sulfanilamide was measured for three independent times to obtain its mass percentages of CHNS elements and then compared with its standard values to obtain the correction factor. The correction factor was updated every 20 measurements, and used for EA calibration during the measurement of real samples. Each real sample was measured four times independently to calculate the mean and deviation. A total of 2 mg was used for each measurement of real sample. The oxidation furnace temperature was 1150 °C and the

reduction furnace temperature was 850 °C. The flow rate of carrier gas (helium) was 200 mL/min and the flow rate of combustion oxygen was 200 mL/min. The combustion time of oxygen addition was 90 s.

NMR measurements were performed on a Bruker BioSpin GmbH instrument at 25 °C. $CDCl_3$ was used as the deuterated solvent. A 50 mg sample was sufficiently mixed with 0.5 mL of $CDCl_3$ and transferred into a 5 mm NMR tube for one-dimensional (1D) liquid $^1$H-NMR characterization. Pulse sequence was zg30 and pulse width was 9.27. Number of scans was 16 and receiver gain was 101. Relaxation delay was 20.0 s and acquisition time was 4.0894 s. Spectrometer frequency was 400 MHz and flip angle was 30°. Acquired size was 32,768 and spectral size was 65,536. The spectral width was 8012.8. The NMR spectra were analyzed via MestReNova software (version 6.1.0-6224 for Windows). The chemical shift of TMS was set as 0 ppm for reference. The n-heptane of AR purity was measured firstly to get the relative ratio of residual solvent (chloroform) signal in deuterated solvent to the TMS signal. The ratio was found to be 0.48 ± 0.04. Based on this ratio and TMS signal, the area for chloroform signal was subtracted and the rest areas integrated for the selected ranges in NMR spectrum were normalized to sum 1 for ease of comparison. Each real sample was measured 3 times independently to calculate the mean and deviation of integrated areas.

For FTIR analysis, the samples were uniformly ground with KBr in an agate mortar and then pressed into round flakes under a pressure of 10 MPa for 60 s. The samples were then measured in transmission mode (TR) on Bruker Tensor II FTIR spectrometry at 25 °C, with a scanning range of 400–4000 cm$^{-1}$ and a resolution of 0.5 cm$^{-1}$. A round flake of pure KBr was scanned for baseline, which was automatically subtracted in each spectrum. Each sample was scanned 16 times and the averaged spectrum was recorded. The infrared (IR) spectrum was processed via the Omnic software (version 8.2.0.387 for Windows).

The XRD analysis was performed with a D8 Advance X-ray diffractometer produced by Bruker (Germany) at 25 °C. The Cu target source (0.15406 nm) was selected. The measurement angle ($2\theta$) ranged from 5° to 60°, which was twice the angle of X-ray incidence ($\theta$). The scan rate was 1°/min and the step size was 0.02°. The spot diameter was 1 mm with LynxEye XE array detector. Asphaltene was ground to fine powder in an agate mortar prior to analysis.

## 3. Results and Discussion

### 3.1. Extractant Compositions Determined via Densimetry

The extractant used in this paper is a binary mixture of xylene (G) and n-heptane (P) in different G/P ratio. For standard samples of extractants with compositions listed in Table 1, their densities $\rho$ were measured and plotted versus the volume fraction $v_G$. $\rho$ shows a linear relation with $v_G$ in Figure 1 and the coefficient of determination ($R^2$) is 0.99967 for the fitting equation

$$\rho = 0.67783 + 0.18164\, v_G, \tag{1}$$

The densities $\rho$ of recovered extractants (P1, P2 and G1, G2) were measured by densimetry and substituted into Equation (1) to get $v_G$. Xylene and heptane of AR purity were used for the 1st extraction of asphaltenes. G1 and P1 were used for the 2nd extraction while G2 and P2 were used for the 3rd extraction. For heptane-rich fractions P1 and P2, their volume fraction of n-heptane $v_P$ ($v_P = 1 - v_G$) is 0.907 ± 0.026 and 0.832 ± 0.011, respectively. The xylene-rich fractions G1 and G2 have higher purity, whose $v_G$ is 0.992 ± 0.014 and 0.966 ± 0.014, respectively, as shown in Figure 1. The negative pressure (vacuum) generated by the pump during vacuum distillation draws away a large amount of solvent and causes a solvent loss and air pollution issue. The average yield for distillate P1 was only 75%. Owing to volatility and good compatibility between xylene and n-heptane, a shorter time of vacuum distillation is preferred to improve the distillate yield and to avoid the xylene vapor dissolving into the heptane-rich fraction. Solvent separation conditions need further optimization, for example, under appropriate negative pressure and distillation time. To reduce the solvent loss in the recovery process, it may be necessary to connect

a cold trap to the vacuum pump or use other separation methods instead of vacuum distillation [22,23]. For blank samples at $v_G = 0$, the $s_B$ of calculated volume concentration is 1.2% and the LOD of $v_G$ is 3.6% for the densimetry method.

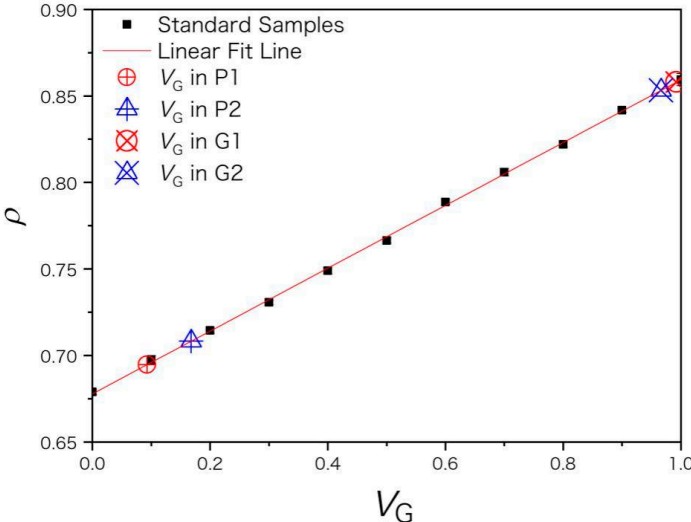

**Figure 1.** The red line is a linear fit of the volume fraction of xylene $v_G$ and density $\rho$ of standard samples (marked in black squares). The real samples of recovered extractants (P1, P2 and G1, G2) are marked by larger symbols along the fitted line. $\rho$ is in unit of g/mL.

The asphaltene yield was 13.2 ± 0.4%, 10.8 ± 0.4% and 8.3 ± 0.4%, respectively, for the 1st, 2nd, and 3rd extractions. The asphaltene yield decreased with the extraction number but the mechanism was not known exactly [7]. To understand the scenarios of asphaltene precipitation more intuitively, the actual volumes of n-heptane and xylene in the solution for settling asphaltenes during the 1st, 2nd, and 3rd extractions were calculated according to their concentrations in Figure 1 and listed in Table 3, assuming 50 mL G and 500 mL P used in each extraction for ease of comparison. With increasing the extraction number, the heptane-rich fraction contains more and more xylenes, reducing the ability of n-heptane to sediment asphaltenes. Simultaneously, the concentration and total volume of xylene in the solution increase. Thus, more asphaltenes are dissolved while the sedimentation amount and final yield decrease. Even when the xylene-rich fraction is as low as $v_G = 96.6\%$, the asphalt can be completely dissolved, while the heptane-rich fraction is still capable to precipitate asphaltene as low as $v_P = 83.2\%$. Thus, the proposed extraction method [7] has good robustness and great prospects for industrialization, since asphaltenes can be efficiently extracted with solvents much lower than AR purity. Goual et al. found that the onset of asphaltene flocculation occurred near a toluene/heptane volume ratio of 70:30 [14]. The xylene/heptane volume ratio is 24:76 for the 3rd extraction in Table 3, well below the critical ratio of asphaltene precipitation. How the precipitation amount varies with the G/P ratio is an interesting question and may help address asphaltene deposits in oil pipelines [9,24].

**Table 3.** The actual volumes of n-heptane (P) and xylene (G) in extractant solution, calculated from the concentration of G and P. 50 mL G and 500 mL P were sequentially added for each extraction. Solvents G and P of AR purity were used for the 1st extraction. G1 and P1 were used for the 2nd extraction while G2 and P2 were used for the 3rd extraction.

| Extraction No. | P (mL) | G (mL) | Total Volume (mL) | G/P Volume Ratio |
|:---:|:---:|:---:|:---:|:---:|
| 1st | 500 | 50 | 550 | 0.10 |
| 2nd | 454 | 96 | 550 | 0.21 |
| 3rd | 418 | 132 | 550 | 0.32 |

### 3.2. Properties of Asphaltenes Extracted at Room Temperature

Asphaltene is usually the most polar and cohesive component of asphalt, and forms the core of the asphalt colloid structure. In Table 4, with increasing the extraction number (from A1 to A3), the mass fractions of C, H and S elements increase while the fractions for element N and others (mainly oxygen) decrease. The fraction of N decreases sequentially from 1.50% in A1 to 1.38% in A3 then to 0.73% in asphalt. Because element N increases the molecular polarity [3], the polarity of asphaltenes from A1 to A3 may decrease with the extraction number and be larger than that of asphalt. C/H atomic ratio is defined as $q$, which decreases sequentially from 0.99 in A1 to 0.78 in asphalt. $q$ reflects the degree of unsaturation of the molecule, usually in range of 0.5 to 1 for bitumen and its components [1,6,7]. $q$ is larger than 1 for fused-ring molecules, and approaches to infinity for chars and graphites [25,26]. As asphaltenes and asphalt have apparently different $q$, their unsaturation discrepancy is much larger than the discrepancy between different asphaltenes [6]. Since the polarity and unsaturation increases from asphalt to A3 then to A1, the π-π conjugation and inter-molecular interactions increase, causing higher cohesion and hardness for asphaltenes than asphalt. Thus, asphaltenes extracted at larger G/P ratio have smaller unsaturation, polarity and inter-molecular interactions.

**Table 4.** Mass percentage of elements from EA (CHNS mode) and the calculated C/H atomic ratio ($q$) for extracted asphaltenes (A1, A2 and A3) and asphalt.

| Fractions | N (%) | C (%) | H (%) | S (%) | Others (%) | $q$ |
|---|---|---|---|---|---|---|
| A1 | $1.50 \pm 0.03$ | $82.46 \pm 0.08$ | $6.95 \pm 0.04$ | $5.97 \pm 0.06$ | $3.13 \pm 0.15$ | 0.99 |
| A2 | $1.44 \pm 0.02$ | $82.62 \pm 0.07$ | $7.07 \pm 0.06$ | $6.09 \pm 0.04$ | $2.79 \pm 0.10$ | 0.97 |
| A3 | $1.38 \pm 0.03$ | $83.31 \pm 0.09$ | $7.24 \pm 0.06$ | $6.20 \pm 0.06$ | $1.87 \pm 0.18$ | 0.96 |
| Asphalt | $0.73 \pm 0.07$ | $83.08 \pm 0.12$ | $8.84 \pm 0.05$ | $4.20 \pm 0.06$ | $3.15 \pm 0.31$ | 0.78 |

Asphaltenes with larger unsaturation are more easily dissolved in solvents with larger G/P ratio, allowing asphaltenes with smaller unsaturation to preferentially precipitate out. Thus, the averaged molecular structure of the extracted asphaltene may change with the extraction number. Owing to complex compositions and chemical structures of asphalt and asphaltenes [5], NMR is generally used to analyze their average molecular structure [1,27] rather than their content quantification [2,28]. As in Figure 2, the protons are divided into A, α, β, and γ categories [1,6,29], whose chemical shifts are, respectively, in ranges of 6.0–9.0 ppm, 2.0–4.0 ppm, 1.0–2.0 ppm and 0.5–1.0 ppm [1,6]. Their integrated areas are denoted as $h_A$, $h_\alpha$, $h_\beta$, and $h_\gamma$, respectively. Protons directly attached to the single (S) and fused (F) rings are in the range of 6.0–7.4 ppm and 7.4–9.0 ppm, respectively [30], whose integral areas are denoted by $h_S$ and $h_F$ in Table 5. From A1 to A3 then to asphalt, both $h_S$ and $h_F$ decrease, which has a consistent trend with $q$ and indicates a decrease of aromatic ring fractions. The aromatic carbon ratio $f_A$ [1] shows a decrease trend while the hydrogen substitution rate around aromatic rings $\sigma$ [6] shows an increase trend. Thus, with increasing the G/P ratio in extractant or with increasing the extraction number, the aromatic rings of extracted asphaltenes have more branched chains and occupy less proportion. From A1 to A3 then to asphalt, $h_A$ decreases while $h_\beta$ and $h_\gamma$ increase. Therefore, the fraction of alkyl branches and (methyl) ends increases in extracted asphaltenes. In terms of averaged molecular structures, the extracted asphaltenes are significantly different from the base asphalt [29]. In terms of physical properties, the asphaltenes obtained from the 1st, 2nd and 3rd extractions are all brittle, hard and shiny, without apparent adhesion to glass, skin or plastics at room temperature, which are apparently different from the base asphalt. The remaining asphalt after asphaltene extraction is much softer and stickier than the original (base) asphalt, which can be used as a new coating material.

**Table 5.** Calculated values for the average molecular structure of the extracted asphaltenes (A1, A2 and A3) and the base asphalt according to their NMR spectra [1].

| NMR Quantities | A1 | A2 | A3 | Asphalt |
|:---:|:---:|:---:|:---:|:---:|
| $h_F$ | $0.08 \pm 0.01$ | $0.06 \pm 0.01$ | $0.04 \pm 0.01$ | $0.02 \pm 0.01$ |
| $h_S$ | $0.11 \pm 0.02$ | $0.09 \pm 0.01$ | $0.08 \pm 0.01$ | $0.04 \pm 0.01$ |
| $f_A$ | $0.59 \pm 0.01$ | $0.56 \pm 0.01$ | $0.54 \pm 0.01$ | $0.40 \pm 0.02$ |
| $\sigma$ | $0.39 \pm 0.02$ | $0.43 \pm 0.01$ | $0.47 \pm 0.02$ | $0.52 \pm 0.03$ |
| $h_A$ | $0.19 \pm 0.02$ | $0.15 \pm 0.01$ | $0.12 \pm 0.01$ | $0.06 \pm 0.01$ |
| $h_\alpha$ | $0.24 \pm 0.03$ | $0.23 \pm 0.01$ | $0.21 \pm 0.01$ | $0.12 \pm 0.02$ |
| $h_\beta$ | $0.43 \pm 0.03$ | $0.47 \pm 0.02$ | $0.49 \pm 0.02$ | $0.62 \pm 0.03$ |
| $h_\gamma$ | $0.13 \pm 0.01$ | $0.14 \pm 0.01$ | $0.17 \pm 0.02$ | $0.20 \pm 0.01$ |

[1] $h_i$ is the normalized integral area (fraction) for protons of $i$ category; $f_A$ is the aromatic carbon ratio and calculated via $f_A = 1 - (h_\alpha + h_\beta + h_\gamma)/(2q)$; $\sigma$ is substitution rate on aromatic rings and calculated via $\sigma = h_\alpha/(2h_A + h_\alpha)$.

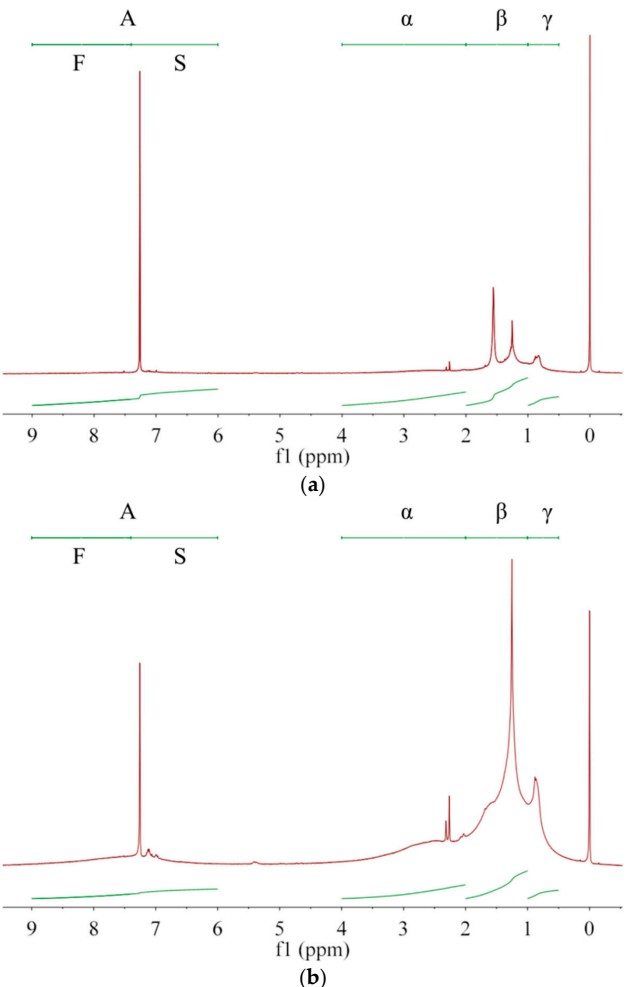

**Figure 2.** *Cont.*

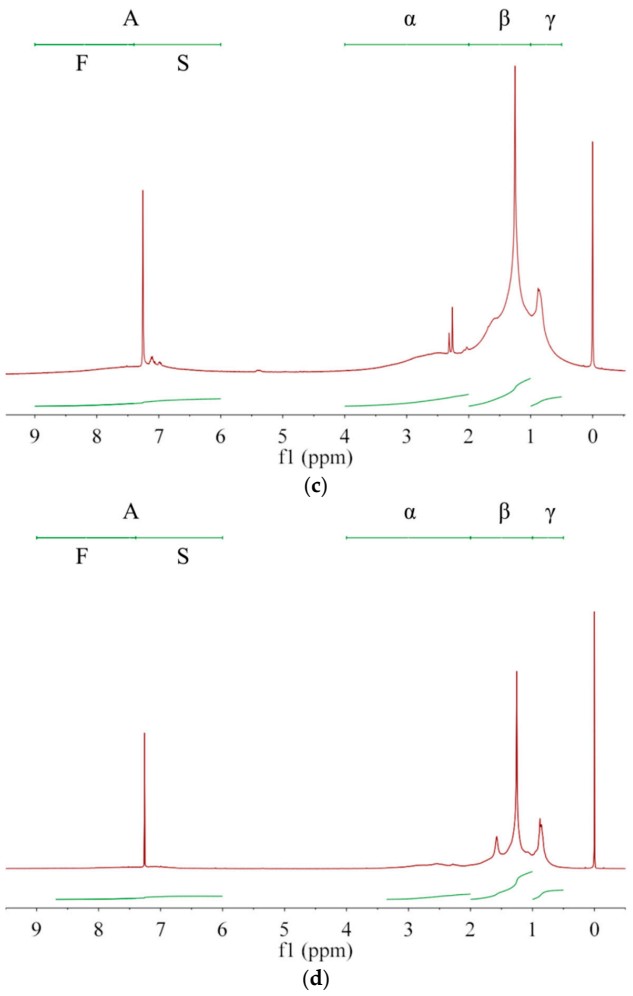

**Figure 2.** The $^1$H-NMR spectra for (**a**) asphaltene A1, (**b**) asphaltene A2, (**c**) asphaltene A3 and (**d**) base asphalt. Protons are divided into A, α, β, and γ categories while the category A includes protons on single (S) and fused (F) aromatic rings.

In Figure 3, main absorption peaks are marked with purple letters on FTIR spectra and the wave numbers for these peaks are listed in Table 6 with peak assignments [6,31,32]. A general rule is that the peak of asphaltenes locates at smaller wave numbers than that of base asphalt. For example, the peak g arises from the skeleton stretching (C=C) of aromatic rings. It locates at 1592 cm$^{-1}$ for A1 and locates at 1603 cm$^{-1}$ for base asphalt. The shift of peak location towards higher wave number from A1 to asphalt arises from the decrease of molecular aromaticity. The aromatic ring makes the distribution of electron cloud density between single-bond and double-bond more uniform. This conjugative effect reduces the electron cloud density of double bond, and reduces its force constant and vibration frequency, so that the peak g moves towards smaller wave number in Figure 3 and Table 6. Moreover, aromatic rings have stronger electron absorption than saturated alkanes. This inductive effect reduces the electron cloud density and reduces the force constant and vibration frequency of the adjacent bond, thus its absorption peak moves towards lower wave number in Table 6. According to the previous EA and NMR results of Tables 4 and 5, from asphalt to A3 then to A1, the values of $h_S$, $h_F$ and $h_A$ increase while the unsaturation and aromaticity of the molecular structure increase. Such increase may improve the inductive and conjugation effects and shift the peak location towards lower wave number, which is consistent with results in Figure 3 and Table 6.

The aromatic rings can only influence a small proportion of chemical bonds adjacent to the rings while asphalt and asphaltene may have dramatically different proportion. In Table 6, the discrepancy between asphaltenes (A1, A2 and A3) is relatively smaller than that

between asphaltenes and base asphalt. For example, the discrepancy of location for peak g is only 3 cm$^{-1}$ between A1 and A3 while it is 11 cm$^{-1}$ between A1 and asphalt, according to Table 6. The reason may be that the proportion of chemical bonds influenced by aromatic rings increases more dramatically (larger discrepancy) from asphalt to asphaltene A3 than it increases from asphaltenes A3 to A1. From asphalt to asphaltene A3, the asphaltene percentage increases from below 20% to nearly 100%. Such a shift of peak location was also observed for asphaltenes and even resins near 1603 and 1030 cm$^{-1}$ in previous FTIR spectra [6]. Moreover, as asphalt was a complex mixture and its four components had significantly different spectra [6], the peak position and height may vary with the proportion of its components. Compared with asphaltenes, the saturates lacked peaks near 1030 cm$^{-1}$ (peak d) and 1600 cm$^{-1}$ (peak g) [6], owing to lacking S and O elements and double-bond in their molecules. The peak around 1377 cm$^{-1}$ is used for the quantification of poly(styrene-*b*-butadiene-*b*-styrene) (AASHTO T302-15, JT/T 1329-2020) or other modifiers in asphalt [31,33,34]. According to Table 6, the position of this peak (e) moves towards a lower wave number from base asphalt to asphaltenes; thus, its specific position may vary with asphaltene proportion and should be checked carefully when calculating peak height or area integral for quantification.

**Table 6.** The wave number (in unit of cm$^{-1}$) for main peaks on IR spectra for the extracted asphaltenes (A1, A2 and A3) and the base asphalt [1].

| IR Peak No. | A1 | A2 | A3 | Asphalt |
|:---:|:---:|:---:|:---:|:---:|
| a | 743 | 743 | 744 | 749 |
| b | 805 | 805 | 807 | 818 |
| c | 857 | 857 | 857 | 861 |
| d | 1027 | 1028 | 1028 | 1032 |
| e | 1372 | 1373 | 1374 | 1377 |
| f | 1452 | 1452 | 1452 | 1457 |
| g | 1592 | 1592 | 1595 | 1603 |
| h | 2849 | 2849 | 2849 | 2853 |
| i | 2919 | 2919 | 2919 | 2924 |
| j | 3053 | 3053 | 3054 | 3055 |

[1.] Peaks a, b and c arise from the (C-H) aromatic ring substitution bands. The peak d may arise from the stretching vibration of sulfoxide (S=O) group. Peaks e and f arise from the bending vibration absorption of aromatic methyl (-CH$_3$) and methylene (CH$_2$) groups. The peak g is the skeleton stretching (C=C) of aromatic rings. The peak h arises from the symmetric stretching vibrations of aliphatic methyl (-CH$_3$) group. The peak i arises from the antisymmetric stretching vibrations of aliphatic methylene (CH$_2$) group. Peak j arises from the stretching vibration (=C-H) on the unsaturated bond or aromatic ring.

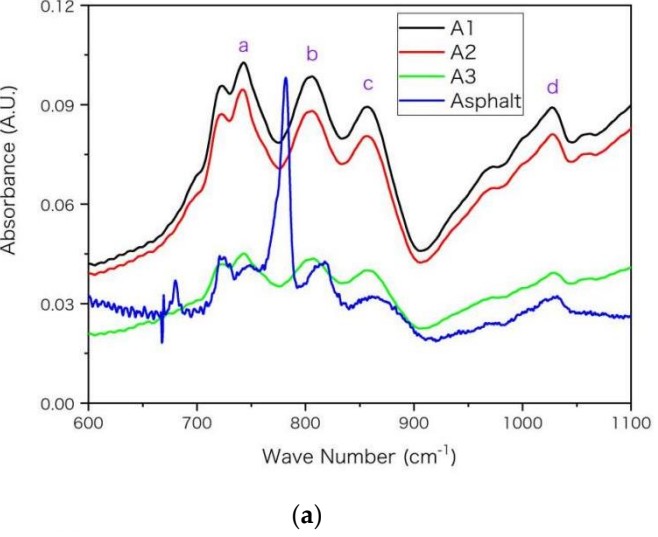

(**a**)

**Figure 3.** *Cont.*

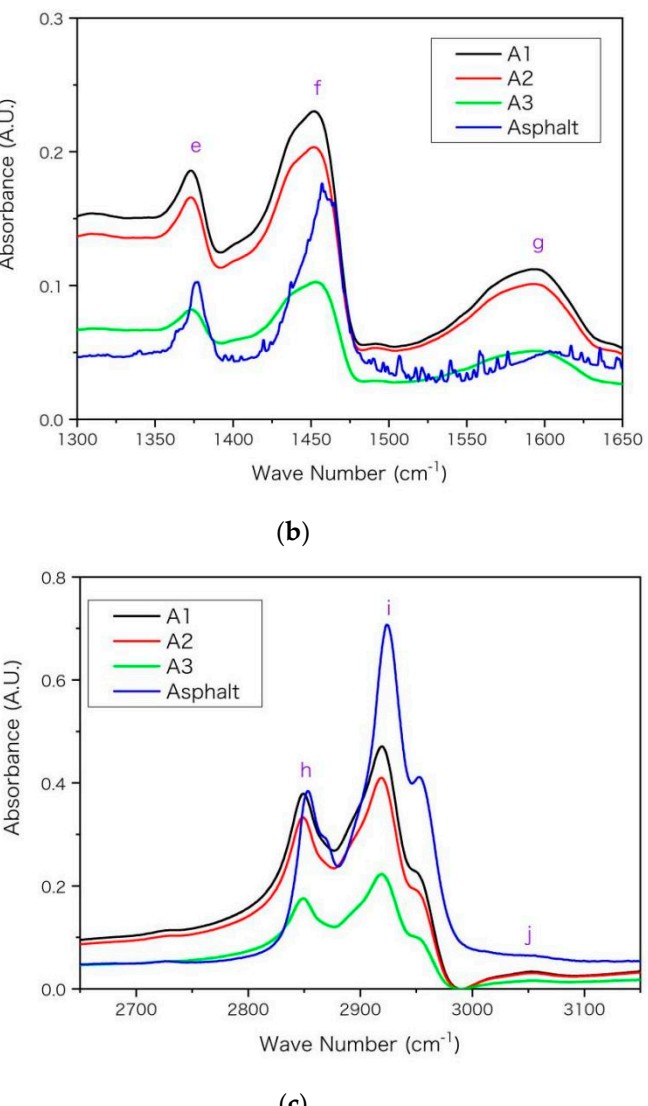

**Figure 3.** The FTIR spectra for the asphaltenes (A1, A2 and A3) and asphalt with wave number (in unit of cm$^{-1}$) in ranges of (**a**) 600–1100, (**b**) 1300–1650, (**c**) 2650–3150, respectively. The absorbance is in arbitrary unit (A.U.) and main peaks are marked in alphabetical order.

The XRD spectra of asphaltenes mainly have 3 characteristic peaks of $\gamma$, 002 and 10 band, which are labelled as $\gamma$, m and a, respectively, in Figure 4. The X-ray wavelength $\lambda$ is 1.5406 angstrom (Å). According to their diffraction angles ($2\theta$) listed in Table 7 and the Bragg equation $d = \lambda/(2\sin\theta)$ [35–37], the characteristic lengths $d$ are calculated and listed in Table 7. The peaks $\gamma$ and m reflect the packing structure of saturated alkyl chains and aromatic rings, respectively, while the peak a is related to the size of the aromatic sheet [35,37,38]. From A1 to A3 in Table 7, the decrease of average distance $d_\gamma$ indicates a denser packing of alkyl chains, while the increase of layer spacing $d_m$ implies a looser packing of aromatic rings. According to previous EA, IR and NMR results, the unsaturation, polarity and aromaticity of asphaltenes decrease while the number of substituted branches and fraction of methyl groups increase from A1 to A3. Thus, the packing of aromatic rings becomes looser owing to more substituted branches on aromatic rings and smaller inter-molecular interactions, while the packing of alkyl chains becomes denser because of the increased fraction of free chain (methyl) end. The decreased ring fraction and molecular unsaturation is also helpful to decrease the molecular rigidity and beneficial to the packing of alkyl chains. Therefore, the variation trend of molecular structure of asphaltenes reflected by XRD spectra is consistent with the previous experimental results.

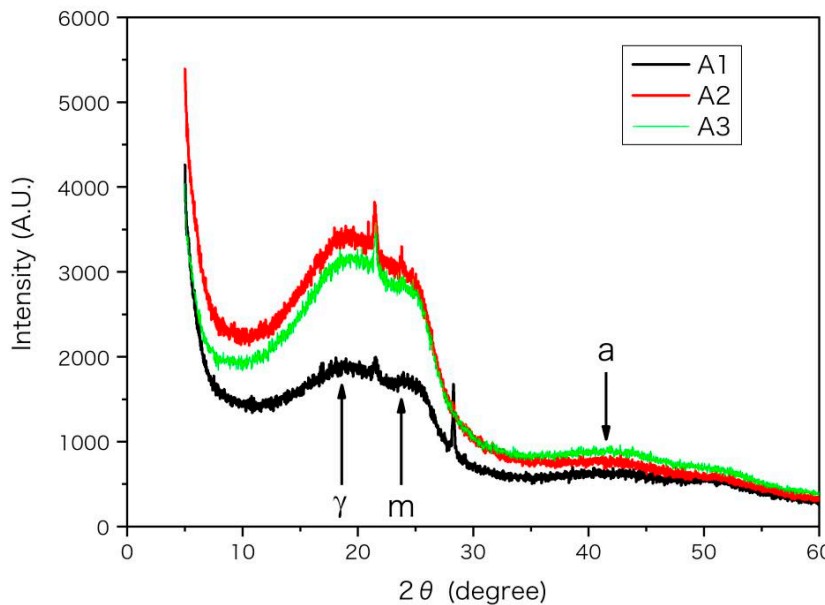

**Figure 4.** The XRD spectra for the asphaltenes (A1, A2 and A3). The diffraction intensity from position sensitive detector (PSD) is in arbitrary unit (A.U.) and plotted versus diffraction angle $2\theta$. The peaks of 002 and 10 band are labelled as m and a respectively.

**Table 7.** The packing length *d* (in unit of Å) calculated according to diffraction angles ($2\theta$) from XRD spectra for the extracted asphaltenes (A1, A2 and A3). Subscripts m and a represent peaks of 002 and 10 band, respectively.

| XRD Quantities | A1 | A2 | A3 |
|---|---|---|---|
| $(2\theta)_\gamma$ | 19.07° | 19.21° | 19.49° |
| $(2\theta)_m$ | 23.90° | 23.79° | 23.76° |
| $(2\theta)_a$ | 41.89° | 40.34° | 41.61° |
| $d_\gamma/\text{Å}$ | 4.650 | 4.617 | 4.551 |
| $d_m/\text{Å}$ | 3.720 | 3.737 | 3.742 |

## 4. Conclusions

The linear equation between the density and concentration for standard samples of extractant containing xylene (G) and n-heptane (P) was fitted with $R^2 = 0.99967$ and used to analyze the real samples of extractant recovered and reused in asphaltene extraction process. With the increase of extraction number, the G/P ratio in extractant increases. The proportion of asphaltenes dissolved in solution rises and the proportion of precipitation from solution falls, resulting in a decrease of the asphaltene yield with the number of extractions. The results of EA, IR, NMR and XRD reached a consistent conclusion. For asphaltenes (A1, A2 and A3) sequentially extracted at room temperature, the fractions of substituted alkyl chains on aromatic rings and chain ends increase while the fractions of N and O elements as well as the C/H atom ratio (unsaturation) decrease, with increasing G/P ratio of extractant or with the extraction number. With increasing the G/P ratio of extractant, the extracted asphaltenes have smaller fractions of C and H of aromatic rings (aromaticity). The decrease of unsaturation and aromaticity from A1 to asphalt leads to reduced conjugative and inductive effects, which shift IR peaks towards higher wave numbers. Since asphaltenes extracted at higher G/P ratio have a smaller fraction of aromatic rings, more substitution rate on aromatic rings and bigger fraction of alkyl and methyl groups in molecular structure; they have smaller molecular rigidity and weaker inter-molecular conjugation. Thus, asphaltenes extracted at a higher G/P ratio have a denser packing of alkyl chains and a looser packing of aromatic rings.

**Author Contributions:** Conceptualization, D.S.; data curation, Y.S.; methodology, D.S.; visualization, Y.S.; writing—original draft, D.S. and Y.S.; writing—review and editing, D.S. and F.C. All authors have read and agreed to the published version of the manuscript.

**Funding:** This research received funding from the Nanxun Collaborative Innovation Center Key Research Project (SYS01001), the Open-ended Fund of Key Laboratory of Urban Pollutant Conversion, Chinese Academy of Sciences (KLUPCKF-2020-4), the Special Research Funds in Shandong Jianzhu University (X20087Z0101) and Doctoral Fund in Shandong Jianzhu University (X20078Z0101).

**Institutional Review Board Statement:** Not applicable.

**Informed Consent Statement:** Not applicable.

**Data Availability Statement:** The data presented in this study and the source codes are available on request from the corresponding author Dachuan Sun.

**Acknowledgments:** The author Dachuan Sun acknowledged Xianglong Zhao and Rui Feng for helpful discussions.

**Conflicts of Interest:** The authors declare no conflict of interest.

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
