# Peer review of "The Influence of Extractant Composition on the Asphaltenes Extracted from Asphalt"

_coatings, doi:10.3390/coatings12101600_

Round 1

Reviewer 1 Report (Previous Reviewer 2)

Earlier, I gave a positive review of the work, because in my opinion the work may be of interest to the readers of the journal. The authors propose a new approach to the analysis of solvent samples after asphaltene extraction, namely the use of 1H NMR. I carefully read the questions of other referees and the answers of the authors. In my opinion, the authors gave answers to the most significant questions and corrected the manuscript. I agree with the removal of the section discussing the use of 13C NMR. In my opinion, in its current form, the work can be accepted for publication.

Author Response

OK, thank you!

Reviewer 2 Report (Previous Reviewer 1)

The manuscript by Dachuan Sun and Yang Song compares the chemical characteristics of 3 different asphaltene fractions extracted from asphalt. The manuscript is generally well written and completely different to the previously submitted paper for measuring solvent ratios. I found that the paper could be of interest to the readers of Coatings, however, there are parts of the NMR and IR experiments that must be improved before the manuscript should be accepted. I therefore recommend that the manuscript be accepted after major revisions have been made.

Major comments:

The chloroform used as an NMR solvent would have a signal around 7.3 ppm and thus overlap with region A. This would need to be subtracted based on the relative ratio of the chloroform signal to the TMS signal. Furthermore, the peaks are very broad leading to significant integration error. Please provide standard deviations for the areas given to allow significant differences between A1, A2 and A3 to be made.

I am confused by what the IR data is showing. The spectra appear to be the same for A1, A2 and A3. There is discussion about another paper’s IR results but no clear explanation of the IR spectra in this manuscript. In the abstract and conclusion there is mention of shifts to the wave numbers to support the difference is conjugation however, there does not appear to be any difference in the wave numbers according to Table 6.

Minor comments:

Please add the full names of techniques before stating the abbreviation

How was the EA calibrated and how much sample was used?

Please add the temperatures at which each characterization method was ran.

The extraction yields are mentioned in section 3.1 but I do not see the values.

Line 258, a reference should be given to support the usual range.

The IR wave numbers should be presented to no decimal places.

Author Response

Reviewer 3 Report (New Reviewer)

After reviewing, it needs to be major revision as follows:

1.Re-title especially "Room Temperature"

2.Redraw "Therefore, asphaltenes extracted at higher G/P ratio have a denser packing of alkyl chains and a looser packing of aromatic rings. Their infrared absorptions generally move towards higher wave numbers owing to smaller aromaticity and weaker conjugation effect."

3.Identify the novelty of "The rest of this paper is organized as follows: the lists of all chemicals and devices, the experimental procedures and characterization methods are shown in Section 2. The densimetry method to determine extractant concentrations and the properties of extracted asphaltenes are shown in Section 3. The conclusions and perspectives are in Section 4"

4.Technical explainations "The RE-52AA rotary evaporator and SHZIII circulating wa- 89 ter vacuum pump are purchased from Shanghai Yarong Biochemical Instrument Factory 90 (Shanghai, China). The filter device with volume 1000 mL is purchased from Jiangsu Feida 91 Glass Products Company (Nanjing, China)."

5.Rewrite "(2) The filtrate is poured into a pear-shaped bottle and recovered by a rotary evaporator. The SHZIII vacuum pump is used to provide vacuum and tap water is used for condensation. The water bath of the rotary evaporator is set to be 45 °C at first and held for 1 h at that temperature to collect the fractions. The solvent condensation is observable from 30 °C and usually completed within 30 min."

6.Recheck "Figure 1. Owing to volatility and good compatibility between xylene and n-heptane, the rotary evaporation time at 40 °C should not be too long, to prevent the xylene vapor from dissolving into the condensates and to improve the purity of heptane-rich fraction. The negative pressure generated by the pump..."

7.Rewrite "On the other hand, the asphalt is soft and sticky, not shiny or brittle. The remaining part of the asphalt after asphaltene extraction is much softer and stickier than the original asphalt, which can be used as a new coating material."

8.Redraw "Among them, saturate had the least number of peaks, lacking peaks near 1030, 1100 and 1600 cm-1 . As asphalt is a complex mixture, the specific position of absorption peak may vary with its compo- 310 nent content. For example, absorption at 1377 cm-1 is deemed as the characteristic peak of asphalt and is used for the quantification of poly(styrene-b-butadiene-b-styrene) (AASHTO T302-15, JT/T 1329-2020) or other modifiers in asphalt [26,28,29]."

9.No need "The future research plan is to study the effect of asphaltene content on bonding strength of asphalt when used as coatings or constructive materials."

Round 2

Reviewer 2 Report (Previous Reviewer 1)

The authors have improved the description of their experiments and results. I recommend that the manuscript be published. I have one comment which is that the term 'physical quantities' in table 5's caption does not seem like the correct choice and could just be removed. The authors should provide formula for how f and sigma are calculated in table 5. 

Author Response

Reviewer 3 Report (New Reviewer)

The revised version can be published.

Author Response

OK, thank you!

This manuscript is a resubmission of an earlier submission. The following is a list of the peer review reports and author responses from that submission.

Round 1

Reviewer 1 Report

The manuscript by Sun and Song compares the use of NMR spectroscopy and densitometry to determine the ratio of heptane and xylene in solvent mixtures. The method was then applied to solvent extraction solutions. I think the results are interesting and that the idea of using NMR is useful. I am not sure this paper will be of interest to people working with coatings. It seems like more of a general analytical or physical chemistry paper. I think more detail in the NMR methodlogy is needed before this paper should be accepted. In its current form I must recommend the manuscript be rejected.

Comments

Proof of quantitative NMR conditions must be shown. What pulse sequences were used? What are the flip angles used? What was the repetition time and was it proven to be longer than 5 times the T1?

NOE enhancement was mentioned as an issue for 13C measurements but if the inverse gated decoupling pulse sequence is used the NOE enhancement is not present. Why was that pulse sequence not tried?

Many statements in the introduction lack references and other papers using NMR for studying solvent mixtures should be cited.

The spectra in Figures 3 and 7 should be more zoomed in on the region of interest with the green lines below the spectra not on top.

Table 7 how were the error bars determined?

Lines 370-371 Please specify if this is the percent of good or bad solvent.

Line 378 Please specify the currency

Reviewer 2 Report

In the manuscript, the authors propose a new method for the analysis of a binary mixture of solvents, viz. isomeric xylenes and n-heptane, using 1H and 13C NMR spectroscopy data. The solvents used contained impurities, so the authors proposed to use of weighted values for the number of C or H atoms, that increased the consistence between the experimental data and the calculated values. The authors show that consistency between the predicted concentrations and concentrations calculated from experimental data are better for 1H NMR-based measurements rather than 13C NMR. This is quite expected due to different relaxation times and NOE enhancement for different carbon atoms. At the same time, the use of 1H NMR data to determine the concentration of solvents in mixtures showed good convergence with densimetry data for the same systems, while the new method demonstrates a number of advantages - it requires a smaller amount of sample for analysis, requires less time for preliminary preparation, and has greater precision and smaller standard deviation. The authors showed the possibility of using the method for the analysis of solvent samples after asphaltene extraction. The results obtained made it possible to propose a mechanism for decreasing the asphaltene yield with increasing extraction time, which is associated with a gradual decrease in the n-heptane fraction in the mixed solvent. The work may be published unchanged in the journal Coatings.

Reviewer 3 Report

The manuscript submitted by Sun and Song is focused on the development of  an NMR method for the quantitative determination of a binary mixture of solvents used for asphaltanes extraction. Despite the applications could be of interest for the coating field, I strongly feel that this paper is too analytical-oriented and should be submitted to a journal more dedicated to method development.

In any case, the manuscript needs an intensive review of its content, as I found very difficult to follow and to understand each section. I think that the authors should make an effort and improve the paper from several points of view:

- the methodology is not clearly described, there is too much attention on using different symbols even when it is not necessary, thus making very difficult for a reader to follow the flow

- the content needs to be more "NMR-like" written, e.g. H atoms are usually named "protons" and some assumptions are well-known and it is not necessary to point them out

- the section about the use of 13C should be completely removed, since as the authors point out the NOE effects make it difficult to use 13C NMR for quantitative determinations

- no information about the NMR parameters used for the analysis is provided, thus making difficult replicating the experiments

- the text reported in the supplementary material should not be the same already present in the main text, the equations could be enough

Round 2

Reviewer 1 Report

The authors have made many changes to the manuscript, however, the two main concerns of the manuscript are not addressed. The repetition time needs to be such that it is longer than 5 times the T1 of all signals integrated. The repetition time here is less than 5 s and thus it is likely not long enough to remove any effect from insufficient relaxation. Secondly, the manuscript does not seem appropriate for the journal and would be better suited in an analytical or physical chemistry journal. 

Reviewer 3 Report

The authors made some improvements in the paper, nevetheless I still find the paper hard to read and to follow. I think that it is too method-oriented and it could rise more interest from an audience focused on NMR methods development.